# Coherent 3D Scene Diffusion
# From a Single RGB Image

**Manuel Dahnert**[1]     **Angela Dai**[1]     **Norman Müller**[2]     **Matthias Nießner**[1]

[1]Technical University of Munich, Germany     [2]Meta Reality Labs Zurich, Switzerland

## Abstract

We present a novel diffusion-based approach for coherent 3D scene reconstruction from a single RGB image. Our method utilizes an image-conditioned 3D scene diffusion model to simultaneously denoise the 3D poses and geometries of all objects within the scene. Motivated by the ill-posed nature of the task and to obtain consistent scene reconstruction results, we learn a generative scene prior by conditioning on all scene objects simultaneously to capture the scene context and by allowing the model to learn inter-object relationships throughout the diffusion process. We further propose an efficient surface alignment loss to facilitate training even in the absence of full ground-truth annotation, which is common in publicly available datasets. This loss leverages an expressive shape representation, which enables direct point sampling from intermediate shape predictions. By framing the task of single RGB image 3D scene reconstruction as a conditional diffusion process, our approach surpasses current state-of-the-art methods, achieving a 12.04% improvement in $AP_{3D}$ on SUN RGB-D and a 13.43% increase in F-Score on Pix3D.

## 1 Introduction

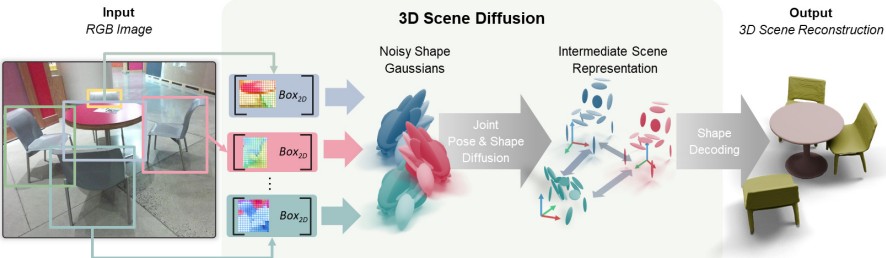

Figure 1: Given a single RGB image of an indoor scene, our model reconstructs the 3D scene by jointly estimating object arrangements and shapes in a globally consistent manner. Our novel diffusion-based 3D scene reconstruction approach achieves highly accurate predictions by utilizing a novel generative scene prior that captures scene context and inter-object relationships, and by employing an efficient surface alignment loss formulation for joint pose- and shape-synthesis.

Holistic 3D scene understanding is crucial for various fields and lays the foundation for many downstream tasks in robotics, 3D content creation, and mixed reality. It bridges the gap between 2D perception and 3D understanding. Despite impressive advancements in 2D perception and 3D reconstruction of individual objects [56, 5, 12, 38], 3D scene reconstruction from a single RGB observation remains a challenging problem due to its ill-posed nature, heavy occlusions, and the

38th Conference on Neural Information Processing Systems (NeurIPS 2024).

complex multi-object arrangements found in real-world environments. While previous works [15, 32, 33] have shown promising results, they often recover 3D shapes independently and thus do not leverage the scene context nor inter-object relationships. This leads to unrealistic and intersecting object arrangements. Additionally, common feed-forward reconstruction methods [48, 77, 37] struggle with heavy occlusions and weak shape priors, resulting in noisy or incomplete 3D shapes, which hinders immersion and hence limits the applicability in downstream tasks. To address these challenges and to advance 3D scene understanding, we propose a novel generative approach for coherent 3D scene reconstruction from a single RGB image. Specifically, we introduce a new diffusion model that learns a generative scene prior capturing the relationships between objects in terms of arrangement and shapes. When conditioned on a single image, this model simultaneously reconstructs poses and 3D geometries of all scene objects. By framing the reconstruction task as a conditional synthesis process, we achieve significantly more accurate object poses and sharper geometries. Publicly available 3D datasets [47, 62] typically only provide partial ground-truth annotations, which complicates joint training of shape and pose. To overcome this, we propose a novel and efficient surface alignment loss formulation $\mathcal{L}_{\text{align}}$ that enables joint training of shape and pose even under the lack of full ground-truth supervision. Unlike previous methods [48, 77] that involve costly shape decoding and point sampling on the reconstructed surface, our approach employs an expressive intermediate shape representation that enables direct point sampling from the conditional shape prior. This provides additional supervision and results in more globally consistent 3D scene reconstructions. Our method not only outperforms current state-of-the-art methods by 12.04% in $\text{AP}^{15}_{\text{3D}}$ on SUN RGB-D [62] and by 13.43% in F-Score on Pix3D [64] but also generalizes to other indoor datasets without further fine-tuning.

In summary, our contributions include:

- A novel diffusion-based 3D scene reconstruction approach that jointly predicts poses and shapes of all visible objects within a scene.

- A novel way for modeling a generative scene prior by conditioning on all scene objects simultaneously to capture scene context and inter-object relationships.

- An efficient surface alignment loss formulation $\mathcal{L}_{\text{align}}$ that leverages an expressive intermediate shape representation for additional supervision, even in the absence of full ground-truth annotation.

## 2 Related Works

The task of 3D scene reconstruction from a single view combines the fundamental domains of 2D perception and 3D modeling into a unified challenge of holistic 3D understanding. Given the multi-faceted nature of the task, we are providing a comprehensive overview of the relevant research directions and contextualizing our contributions.

### 2.1 Single-View 3D Reconstruction

**Object Reconstruction.** Since the foundational work by Roberts [54], numerous methods have been developed to learn cues for deriving 3D object structures, thereby bridging the gap between 2D perception and the 3D world. These methods typically involve an image encoder network that processes the input image of a single object, capturing its features. The extracted features are either correlated with an encoded shape database to retrieve a suitable shape [32, 33, 17], or used by a 3D decoder to reconstruct the object in a specific 3D representation, such as voxel grids [8, 72], point clouds [14, 43], meshes [70, 66], or neural fields [73, 27]. [19] uses a message-passing graph network between geometric primitves to reason about the structure of the shape.

**Scene Reconstruction.** Early works formulated single-view scene reconstruction as 3D scene completion from given or estimated depth information [63, 10, 78, 9] in a volumetric grid. While these methods have produced promising results, their representational power to model fine details is limited by the spatial resolution of the 3D grid. Multi-object reconstruction and scene parsing methods represented objects using primitives [13, 23], voxel grids [68, 35, 52], or CAD models [26, 24], while also considering the relation between the objects [31]. The approach presented by Nie *et al.* [48] is particularly relevant, proposing a holistic method for joint pose and shape estimation from a single image. Zhang *et al.* [77] extended this idea by incorporating an implicit shape representation and

an additional pose refinement using a graph neural network. Although these methods provided significant advances in holistic scene understanding, they struggled with accurate pose estimation and produced noisy scene objects, leading to intersecting or incomplete objects. In contrast to these previous works, we are proposing a generative method to obtain a strong scene prior and formulate the reconstruction task as a conditional synthesis task. This allows for more robust reconstruction that is less prone to object insections or implausible object geometries.

## 2.2 3D Diffusion Models

In recent years, denoising diffusion probabilistic models (DDPMs) have emerged as a versatile class of generative models, demonstrating impressive results in image and video generation. Unlike other classes of generative models such as auto-regressive models [46, 75, 59], Generative Adversarial Networks (GANs) [71, 79] and Variational Autoencoders (VAEs), diffusion models iteratively reverse a Markovian noising process. This method ensures stable training and has the ability to capture diverse modes while producing detailed outputs. Several approaches have utilized diffusion models to learn the distribution of individual 3D shapes using various 3D representations, including volumetric grids [6, 7, 25], point clouds [42, 74], meshes [2], implicit functions [30], neural fields [45, 58, 29] or hybrid representations [80, 76]. [53] propose a hierarchical voxel diffusion model, which is capable of modelling large-scale and fine-detailed geometry. While these methods can synthesize high-quality 3D shapes, they typically focus on single objects in canonical space. In contrast, we are proposing a diffusion-based approach that addresses the more challenging problem of multi-object scene reconstruction, encompassing accurate pose estimations and an understanding of inter-object relationships.

**Conditional Diffusion for 3D Reconstruction.** Recent works also use diffusion models for single-view object reconstruction [6, 7, 44]. For instance, [65] learns the shape distribution of a single category by denoising a set of 2D images for each object, while [44] projects image features onto noisy point clouds during the diffusion process to ensure geometric plausibility. Recently, several works proposed to leverage multi-view consistency within pre-trained text-conditional 2D image diffusion models to reconstruct individual 3D objects [38, 51, 57]. Similar to our work, Tang *et al.* [67] use a diffusion model to learn scene priors from synthetic data, showing unconditional scene synthesis of a single room type and text-conditional generation. However, their approach does not support image-based scene reconstruction. Furthermore, it depends on clean synthetic data, which provides full 3D ground truth supervision and CAD model retrieval, thereby limiting shape diversity. While these existing methods have shown promising results on single objects or synthetic scenes, our approach targets real-world scenes. By framing the reconstruction task as a conditional generation process, our scene prior accurately delivers poses and shapes of multiple objects, even in the presence of strong occlusions, significant clutter, and challenging lighting conditions.

## 3 Method

### 3.1 Overview

Our method takes a single RGB image of an indoor scene as input and generates a globally consistent 3D scene reconstruction that matches the input image. To this end, we are framing the reconstruction task as a conditional generation problem using a diffusion model conditioned on the input view (Sec. 3.2), which simultaneously predicts the poses (Sec. 3.3) and shapes (Sec. 3.4) of all objects in the scene. Given the ill-posed nature of single-view reconstruction, such a probabilistic formulation is particularly well-suited for this task. To ensure accurate reconstructions and to learn a strong scene prior, we model inter-object relationships within the scene using an intra-scene attention module (Sec. 3.5). Additionally, recognizing the incomplete ground truth in many 3D indoor scene datasets, we introduce a loss formulation for joint shape and pose training, which enables training under only partially available supervision (Sec. 3.6). An overview of our approach is illustrated in Fig. 1. In the following sections, we describe each individual contribution in more detail.

### 3.2 Conditional 3D Scene Diffusion

We frame the scene reconstruction task as a conditional generation process via a diffusion formulation [22]. Given an instance-segmented RGB image $\mathbf{I}$ containing a variable number of 2D objects $b_i$

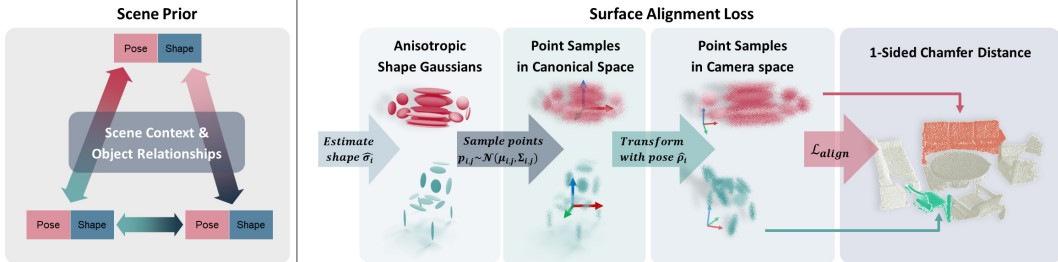

Figure 2: **Scene Prior and Surface Alignment Loss Overview.** (Left) We propose a novel way to model scene priors (Sec. 3.5) by modeling the scene context and the relationships between all objects during the denoising process. (Right) For additional supervision and joint training, we use a surface alignment loss (Sec. 3.6) between a given ground truth depth map and point samples directly drawn from the intermediate shape representation $\hat{\sigma}_i$ and transformed to camera space with the predicted object pose $\hat{\rho}_i$.

for $i \in \{1, \ldots, n\}$, our model $\mathbf{\Phi}$ simultanteously estimates all 3D objects $\mathbf{o_i} = (\rho_i, \sigma_i)$ with 7-DoF poses $\rho_\mathbf{i}$ and 3D geometries $\sigma_\mathbf{i}$:

$$(\hat{\mathbf{o}}_1, \ldots, \hat{\mathbf{o}}_n) = \mathbf{\Phi}(\mathbf{I}|(\mathbf{b}_1, \ldots, \mathbf{b}_n)). \tag{1}$$

During the *forward process*, we gradually add Gaussian noise to a data point $x_0$ to $x_T$ over a series of discrete time steps $T$. For a given data point $x_0$, *e.g.*, shapes $\sigma_i$ and poses $\rho_i$, the noisy version $x_t$ at time step $t$ is given by a Markovian process [22, 60] $q(x_t|x_{t-1})$ and its joint distribution $q(x_{1:T}|x_0)$ can be expressed as:

$$q(x_t|x_{t-1}) = \mathcal{N}(x_t; \sqrt{1-\beta_t}x_{t-1}, \beta_t\mathbf{I}), \tag{2}$$

$$q(x_{1:T}|x_0) = \prod_{i=1}^{T} q(x_t|x_{t-1}) \tag{3}$$

with $t \in [1, T]$ and $\beta_t$ a pre-defined linear variance schedule.

During the *reverse process*, the denoising network $\Phi$ tries to remove the noise and recover $x_0$ from $x_T$ as $p_\Phi(x_{t-1}|x_t, y)$

$$p_\Phi(x_{t-1}|x_t, y) = \mathcal{N}(x_{t-1}; \mu_\Phi(x_t, t, y), \Sigma_\Phi(x_t, t, y)), \tag{4}$$

$$p_\Phi(x_{0:T}|y) = p_\Phi(x_T) \prod_{t=1}^{T} p_\Phi(x_{t-1}|x_t, y) \tag{5}$$

with $y$ being the conditional information from the input image $\mathbf{I}$.

**Conditioning.** To effectively guide the diffusion process $p_\Phi(x_{0:T}|y)$, it is crucial to accurately model the conditional information $y$. First, we encode the input image $\mathbf{I}$ using a 2D backbone $\Theta_I$ and apply 2D instance segmentation to get $n$ detected 2D objects $b_i$, comprising of its 2D bounding box, image feature patch, and semantic class (cls). Each element is encoded using a specific embedding function $\Theta$. The per-instance $y_i$ and scene condition $y$ is then formed as:

$$y_i = \text{concat}(\Theta_{\text{box}}(\text{box}_i), \Theta_{\text{feat}}(\text{feat}_i), \Theta_{\text{cls}}(\text{cls}_i)), \tag{6}$$

$$y = (y_1, \ldots, y_n). \tag{7}$$

To learn a scene prior over all objects in the scene, we condition the denoising network on the scene condition $y$. This not only enables learning the individual object representations $o_i$ but also facilitates learning to capture the scene context and inter-object relationships (Sec. 3.5). Furthermore, we adopt classifier-free guidance [21] for our model by dropping the condition $y$ with probability $p = 0.8$, *i.e.*, using a special 0-condition $\varnothing$. This allows our model to function as a conditional model $p_\Phi(x_0|y)$ and unconditional model $p_\Phi(x_0)$ at the same time, thus enabling unconditional synthesis (Appendix B).

**Loss Formulation.** Unlike related works like [23, 48, 77] that regress object poses $\rho_{\mathbf{i}}$ and shape parameters $\sigma_{\mathbf{i}}$ using a multitude of highly-tuned losses, we train our model $\Phi$ to minimize simple diffusion and alignment losses:

$$\mathcal{L}_{\text{joint}}(\mathbf{I}) = \mathcal{L}_{\text{pose}}(\mathbf{I}) + \mathcal{L}_{\text{shape}}(\mathbf{I}) + \lambda \mathcal{L}_{\text{align}}, \tag{8}$$

$$\mathcal{L}_{\text{pose}}(\mathbf{I}) = \mathbb{E}_{\epsilon \sim \mathcal{N}(0,1), t} \| \hat{\epsilon}_{\rho}(\tilde{\rho}(t), t, \mathbf{I}, \mathbf{b}) - \epsilon \|, \tag{9}$$

$$\mathcal{L}_{\text{shape}}(\mathbf{I}) = \mathbb{E}_{\epsilon \sim \mathcal{N}(0,1), t} \| \hat{\epsilon}_{\sigma}(\tilde{\sigma}(t), t, \mathbf{I}, \mathbf{b}) - \epsilon \|, \tag{10}$$

where we define $\tilde{z}(t) = \sqrt{\bar{\alpha}_t} z + \sqrt{1 - \bar{\alpha}_t} \epsilon$ for $z \in \{\rho, \sigma\}$ with pre-defined noise coefficients $\bar{\alpha}_t$, while $\hat{\epsilon}_z$ denotes the predicted noise. We use $\lambda = 0.01$ to balances the effect of $\mathcal{L}_{\text{align}}$.

Due to the lack of full ground truth supervision in publically available 3D datasets, we introduce an additional alignment loss $\mathcal{L}_{\text{align}}$ for joint training of pose and shape (Sec. 3.6). Depending on the availability of ground-truth data (see Sec. 4.2, we mask out individual losses.

### 3.3 Object Pose Parameterization

We adopt the object pose parameterization of [23], defining the pose $\rho_i = (c_i, s_i, \theta_i)$ of an object by its 3D center $c_i \in \mathbb{R}^3$, the spatial size $s_i \in \mathbb{R}^3$, and orientation $\theta_i \in [-\pi, \pi)$ in . The 3D center $c_i$ is further represented by the 2D offset $\delta_i \in \mathbb{R}^2$ between the 2D bounding box center coordinate and the projected coordinate of the 3D center on the image plane, along with the distance $d_i \in \mathbb{R}$ from the object center to the projected center. Our model learns to denoise this 7-dim. pose representation.

### 3.4 Shape Encoding

We represent object shapes using the disentangled shape representation from [20]. A shape is represented as a shape code $\sigma_i \in \mathbb{R}^{256}$ which is factorized into a set of $g$ oriented, anisotropic 3D Gaussians $G_j, j \in \{1, ..., g\}$ and an associated 512-dim. latent feature vector per Gaussian. Each Gaussian consist of 16 main parameters: $\mu_j \in \mathbb{R}^3$ (center), factorized covariance matrix $U_j \in \mathbb{R}^{3 \times 3}$ (rotation), $\lambda_j \in \mathbb{R}^3$ (scale) and $\pi_j \in \mathbb{R}^1$ ("mixing" weight). We use $g = 16$ Gaussians to form a scaffolding of the shape's geometry. Together with their latent features, these Gaussians are decoded into high-fidelity occupancy fields, and the final mesh is extracted by applying marching cubes [40].

While similar to [30], our model learns to denoise this shape parameterization $\sigma_i$, our additional surface alignment loss $\mathcal{L}_{\text{align}}$ (Sec. 3.6) provides relational signal between predicted shapes and poses. This enables additional guidance in the face of missing joint pose and shape annotations as in SUN RGB-D dataset [62].

### 3.5 Scene Prior Modeling

Given the ill-posed nature of single-view reconstruction, a robust scene prior is essential for achieving good performance. Effectively capturing the scene context and modeling the relationships between objects within the scene is crucial for learning this strong scene prior [31, 77]. Previous methods either reconstruct each object individually [15] or refine their features using graph networks [77]. In contrast, our approach considers the entire scene by conditioning on all scene objects simultaneously $p_{\Phi}(x_0|y)$ and $y = (y_1, \ldots, y_N)$ and additionally allows objects to exchange relational information throughout the entire process. We model the inter-object relationships using an attention formulation [69], which has proven to be powerful for aggregating contextual information.
We denote this formulation as Intra-Scene Attention (ISA), which allows all objects within the scene to attend to each other, effectively modeling their relationships. Please refer to Appendix E for more details and to Tab. 2 for the corresponding ablation study, which demonstrates the effectiveness of our learned scene prior.

### 3.6 Surface Alignment Loss

Publically available 3D scene datasets often only provide partial ground-truth annotations [47, 62]. To facilitate joint training of our model on pose and shape estimation, even in the absence of complete ground-truth annotations, we propose to leverage our expressive intermediate shape representation

to provide additional supervision and to align shapes efficiently with the available partial depth information $\mathcal{D}$. An illustration of the surface alignment loss formulation is provided in Fig. 2.

During training, for each object $o_i$, we use the expected shape code $\hat{\sigma}_i$ estimation by our model to obtain the predicted Gaussian $\hat{G}_{i,j}$ distribution. Given this scaffolding representation, we directly sample $m = 1000$ points $p_{(j,l)} \sim \mathcal{N}(\mu_j, \Sigma_j)$ per Gaussian $\hat{G}_{i,j}$ resulting in a shape point cloud $P_i = \{p_{(j,l)} | j \in \{1, \ldots, g\}, l \in \{1, \ldots, m\}\}$. We transform the resulting shape points $P_i$ into the camera frame by the predicted object pose $\hat{\rho}_i$. Using the instance segmentations and ground-truth depth maps, we obtain $K_i$ surface points $q_k^i$ for object $o_i$ and define the surface alignment loss for all visible objects as 1-sided Chamfer Distance [16, 48]

$$\mathcal{L}_{\text{align}} = \frac{1}{n} \sum_{i=1}^{n} \frac{1}{K_i} \sum_{k=1}^{K_i} \min_{p \in P_i} \|q_k^i - p\|_2^2. \tag{11}$$

Unlike previous works such as [48] that perform costly sampling of points on the decoded shape surface, our approach enables direct point sampling from the conditional shape prior $\hat{G}_{i,j}$. This loss formulation facilitates joint training of pose and shape for all objects simultaneously and its efficacy is demonstrated through ablation studies in Tab. 2.

### 3.7 Architecture

Our architecture consists of a pre-trained image backbone, a novel image-conditional scene prior diffusion model, and a conditional shape decoder diffusion module. We utilize an off-the-shelf 2D instance segmentation model, Mask2Former [5], which is pre-trained on COCO [36] using a Swin Transformer [39] backbone, to obtain instance segmentation and image features. Please refer to Appendix E for details about the condition embedding functions.

To denoise object poses $\rho_i$, we use a 1-dim. UNet [55] architecture with 8 encoding and decoding blocks with skip connections. Each block consists of a time-conditional ResNet [18] layer, multi-head attention between the per-object condition $y_i$ and the pose representation, and our intra-scene attention module (Sec. 3.5) to enable relational information exchange and effectively train a scene prior. We use 8 attention heads, with 64 features per head.

To estimate object shapes $\sigma_i$ from the input view $\mathbf{I}$, we denoise the unordered set of Gaussian $G_{i,j}$ using a Transformer [69] model with 2 encoder layers, 6 decoder layers, and multi-head attention with 4 heads to the object condition information, similar to [30]. The per-Gaussian latent features are denoise with a shape decoder diffusion model, realized as another Transformer model with 6 encoder and decoder layers, which is conditioned on the shape Gaussians.

### 3.8 Training and Implementation Details

For all diffusion training processes, we uniformly sample time steps $t = 1, \ldots T, T = 1000$, and use a linear variance schedule with $\beta_1 = 0.0001$ and $\beta_T = 0.02$. We implement our model in PyTorch[50] and use the AdamW [41] optimizer with a learning rate of $1 \times 10^{-4}$ and $\beta_1 = 0.9, \beta_2 = 0.999$. We train our models on a single RTX3090 with 24GB VRAM for 1000 epochs on Pix3D, for 500 epochs on SUN RGB-D and for 50 epochs of additional joint training using $\mathcal{L}_{\text{align}}$.

During inference, we employ DDIM [61] with 100 steps to accelerate sampling speed. For classifier-free guidance [21], we drop the condition $y$ with probability $p = 0.8$.

## 4 Experiments

In the following sections, we will demonstrate the advantages of our method and contributions by evaluating it against common 3D scene reconstruction benchmarks.

### 4.1 Baseline Methods

We compare our method against current state-of-the-art methods for holistic scene understanding: Total3D [48], Im3D [77], and InstPIFu [37]. Total3D [48] directly regresses 3D object poses from image features and uses a mesh deformation and edge-removal approach [49] to reconstruct a shape. Im3D [77] utilizes an implicit shape representation and a graph neural network to refine the pose

predictions. InstPIFu [37] focuses on single-object reconstruction and proposes to query instance-aligned features from the input image in their implicit shape decoder to handle occlusion. For scene reconstruction, they rely on the predicted 3D poses of Im3D. We use the official code and checkpoints provided by the authors of these baseline methods and evaluate with ground truth 2D instance segmentation and camera parameters to ensure a fair comparison. We further compare against a retrieval-based method, ROCA [17] in Appendix D.

## 4.2 Datasets

Following [23, 48, 77], we train and evaluate the performance of our 3D pose estimation on the SUN RGB-D [62] dataset with the official splits. This dataset consists of 10,335 images of indoor scenes (offices, hotel rooms, lobbies, furniture stores, etc.) captured with four different RGB-D cameras. Each image is annotated with 2D and 3D bounding boxes of objects in the scene. During joint training, we use the provided depth maps together with instance masks to compute $\mathcal{L}_{\text{align}}$.
We train and evaluate the performance of our 3D shape reconstruction on the Pix3D [64] dataset, which contains images of common furniture objects with pixel-aligned 3D shapes from 9 object classes, comprising 10,046 images. We use the train and test splits defined in [37], ensuring that 3D models between the respective splits do not overlap.

## 4.3 Evaluation Protocol

For quantitative comparison against baseline methods, we follow the evaluation protocol of [48]. For pose estimation, we report the intersection over union of the 3D bounding box ($\text{IoU}_{\text{3D}}$) and average precision with an $\text{IoU}_{\text{3D}}$ threshold of 15% ($\text{AP}_{\text{3D}}^{15}$) on the SUN RGB-D dataset [62]. In line with previous works [48, 77], we evaluate with oracle 2D detections but also provide camera parameters to all methods during evaluation. To further assess the alignment of the 3D shapes in the scene, we calculate $\mathcal{L}_{\text{align}}$ between reconstructed shapes and the instance-segmented ground-truth depth map. For single-view 3D shape reconstruction, we follow evaluate on the Pix3D [64] dataset. We follow [37] and sample 10,000 points on the predicted shape surface, extracted with Marching Cubes [40] at a resolution of $128^3$, and on the ground truth shapes and evaluate Chamfer distance (CD $\times 10^3$) and F-score after mesh alignment.

## 4.4 Comparison to State of the Art

**3D Scene Reconstruction.** In Fig. 3, we present qualitative comparisons of our approach against state-of-the-art methods for single-view 3D scene reconstruction. The results from Total3D often exhibit intersecting objects and lack global structure. Additionally, their deformation and edge-removal approach results in 3D shapes with visible artifacts and limited details. While the implicit shape representation of Im3D is more flexible, it often produces incomplete and floating surfaces. In contrast, our diffusion-based reconstruction method, as shown in Tab. 1, learns strong scene priors, resulting in a +0.2 improvement in $\mathcal{L}_{\text{align}}$ and more coherent 3D arrangements of the objects in the scene (+12.04% $\text{AP}_{\text{3D}}^{15}$), as well as high-quality and clean shapes (+13.43% F-Score).
Furthermore, we demonstrate the generalizability of our model to other indoor datasets. We evaluate our approach on individual frames from the ScanNet [11] dataset using 2D instance predictions from Mask2Former without additional fine-tuning. As shown in Fig. 4, our method accurately reconstructs the given input view with matching poses and high-quality 3D geometries.
In Appendix D, we additionally train on ScanNet and compare against ROCA [17]. Due to its retrieval approach, the shapes are complete. However, the resulting quality can limited by the diversity of the shape database, which can lead to suboptimal results, see Fig. 11.

**3D Pose Estimation & Scene Arrangement.** As shown in Tabs. 1 and 6, our method outperforms all baseline methods by a significant margin in terms of $\text{IoU}_{\text{3D}}$ and $\text{AP}_{\text{3D}}^{15}$, *i.e.*, improving $\text{mAP}_{\text{3D}}^{15}$ by 12.04% over Im3D [77]. Detailed per-class results are provided in Tabs. 6 and 8. Figs. 3 and 7 demonstrate that our approach effectively learns common object arrangements, such as multiple chairs surrounding a table, while ensuring that furniture pieces do not intersect or float in the air. We attribute these improvements to our model's robust scene understanding, which is derived from learning a strong scene prior that accounts for inter-object relationships.

Table 1: **Quantitative evaluation of 3D scene reconstruction on SUN RGB-D [62] (left) and 3D shape reconstruction on Pix3D [64] (right).** Our 3D scene diffusion approach outperforms all baseline methods on both tasks on common 3D scene reconstruction metrics.

| | SUN RGB-D [62] | | | | | Pix3D [64] | | | |
| --- | --- | --- | --- | --- | --- | --- | --- | --- | --- |
| | $IoU_{3D}$ ↑ | | $AP^{15}_{3D}$ ↑ | | $\mathcal{L}_{align}$ ↓ | | CD ↓ | | F-Score ↑ | |
| Total3D [48] | 20.52 | (-15.58) | 30.56 | (-27.62) | 1.35 | (-0.36) | 44.32 | (-29.27) | 36.20 | (-22.51) |
| Im3D [77] | 28.31 | (-7.79) | 46.14 | (-12.04) | 1.24 | (-0.25) | 51.31 | (-36.26) | 21.45 | (-37.26) |
| InstPIFu [37] | 26.14 | (-9.96) | 45.02 | (-13.16) | 1.19 | (-0.20) | 24.65 | (-9.6) | 45.28 | (-13.43) |
| Ours | **36.10** | | **58.18** | | **0.99** | | **15.05** | | **58.71** | |

Table 2: **Ablations.** We ablate the effect of our contributions and design decisions. We observe significant gains by introducing our proposed scene prior and intra-scene attention module, using denoising diffusion compared to regression, and jointly training shape and pose together.

| Diffusion | ISA | Joint | $IoU_{3D}$ ↑ | | $AP^{15}_{3D}$ ↑ | | $\mathcal{L}_{align}$ ↓ | |
| --- | --- | --- | --- | --- | --- | --- | --- | --- |
| ✗ | ✓ | ✗ | 28.98 | (-7.12) | 47.10 | (-11.08) | 1.18 | (-0.19) |
| ✓ | ✗ | ✗ | 28.82 | (-7.28) | 48.88 | (-9.30) | 1.12 | (-0.13) |
| ✓ | ✓ | ✗ | 35.16 | (-0.94) | 56.07 | (-2.11) | 1.06 | (-0.07) |
| ✓ | ✓ | ✓ | **36.10** | | **58.18** | | **0.99** | |

**3D Object Reconstruction.** In Tab. 1, we quantitatively compare the single-view shape reconstruction performance of our approach against baseline methods on the Pix3D dataset. The results demonstrate that modeling single-view reconstruction as conditional generation over a robust shape prior leads to significant improvements in Chamfer Distance (+9.6%) and F-Score (+13.43%). Detailed per-class results can be found in Tabs. 7 and 9. Fig. 9 illustrates that InstPiFU often reconstructs noisy and incomplete shapes. In contrast, our approach produces clean 3D geometries with fine details, such as thin chair legs and the crease between pillows of a sofa.

In Fig. 5, we show unconditional results by injecting $\varnothing$ as a condition (Sec. 3.2), showcasing that our shape prior models detailed and diverse shape modes across several semantic classes. In Fig. 10, we additionally visualize the shape decomposition capabilities resulting from our shape encoding and the scaffolding Gaussian representation.

## 4.5 Ablations Studies

We conduct a series of detailed ablation studies to verify the effectiveness of our design decisions and contributions. The quantitative results are provided in Tab. 2.

**What is the effect of the denoising formulation?** To assess the benefits of the denoising diffusion formulation, we construct a 1-step feed-forward regression model that uses the same conditional information as input features and model architecture but regresses the object outputs directly in a single timestep. As shown in Tab. 2, modeling 3D scene reconstruction as a conditional diffusion process, rather than using a feed-forward regression formulation, results in significant improvements of +11.08% $AP^{15}_{3D}$ and +0.19 $\mathcal{L}_{align}$.

**What is the effect of our scene prior modeling?** We evaluate the impact of learning a scene prior by modeling the distribution of all objects and their relationships compared to learning the marginal per-object distribution, *i.e.*, predicting each object individually. As shown in Tab. 2, our joint-object scene prior yields a significant improvement of +9.30% $AP^{15}_{3D}$ over per-object prediction. This improvement underscores the importance of learning a robust scene prior that effectively captures inter-object relationships.

**What is the effect of joint training?** We investigate the benefit of joint training for pose and shape using $\mathcal{L}_{align}$ compared to individual training of pose estimation and shape reconstruction. Although our model already learns strong scene and shape priors, Tab. 2 shows that joint training provides additional benefits, resulting in an improvement of +2.11% in $AP^{15}_{3D}$ and +0.07 in $\mathcal{L}_{align}$.

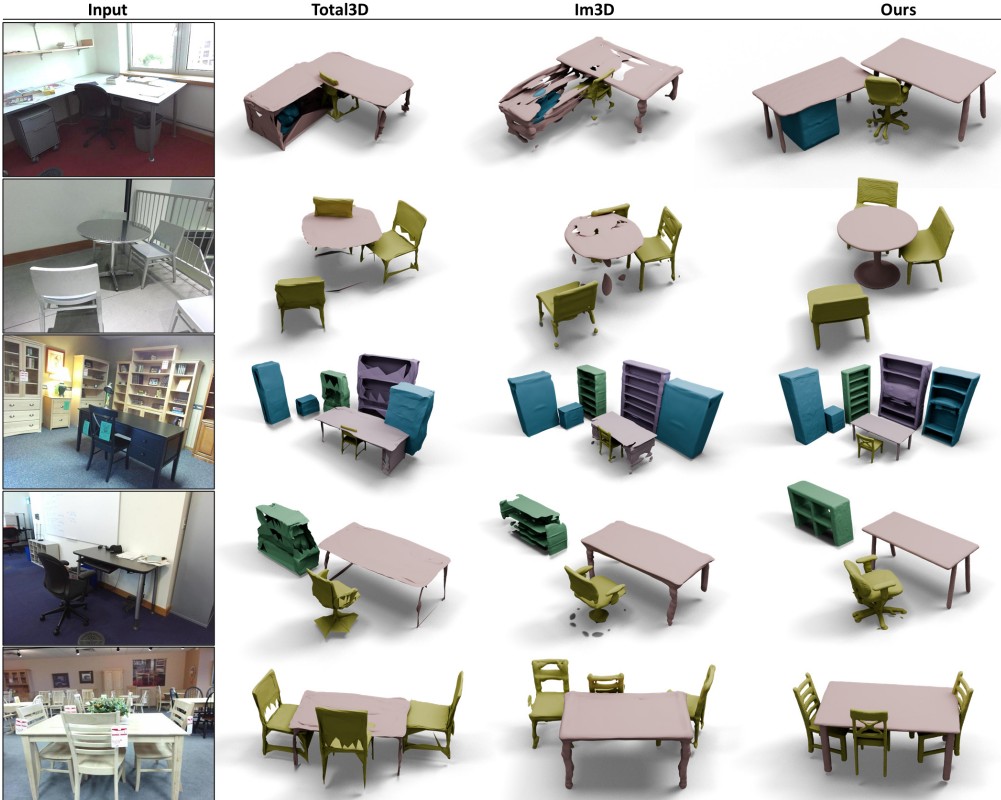

Figure 3: **Qualitative comparison of 3D scene reconstruction on SUN RGB-D [62].** While the baselines often produce noisy or incomplete shape reconstruction of intersecting or misplaced objects, our method produces plausible object arrangements as well as high-quality shape reconstructions.

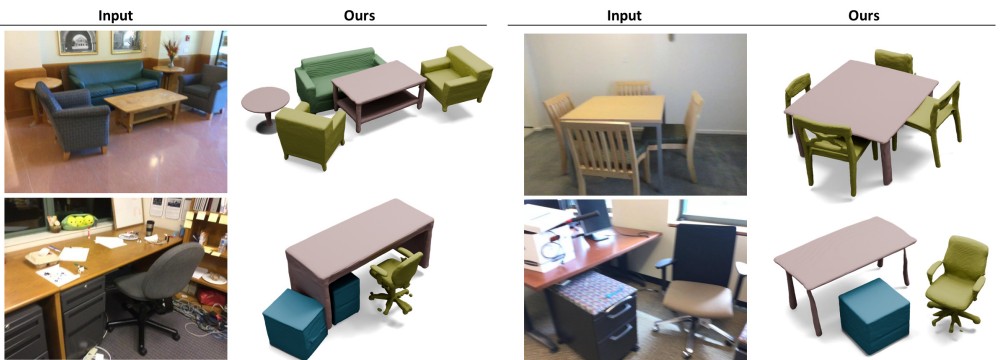

Figure 4: **Inference results on ScanNet [11].** We use our model trained on SUN RGB-D [62] and perform inference on individual frames of ScanNet without fine-tuning. We observe strong generalization capabilities with respect to different camera parameters and scene arrangements.

## 4.6 Limitations

While our conditional scene diffusion approach for single-view 3D scene reconstruction demonstrates significant improvements, there are some limitations. First, our method relies on accurate 2D object detection, making it dependent on the performance of 2D perception models. Upcoming state-of-the-art 2D detection models [1] can be seamlessly integrated to enhance the performance of our approach. Second, our shape prior, trained on a diverse set of semantic classes using 3D shape supervision, does not generalize to unseen object categories. This can be mitigated by combining our model for known categories with single-object diffusion models that leverage pre-trained text-image generation models for 3D shape synthesis [38] of uncommon shape categories. While accurate 3D scene reconstruction

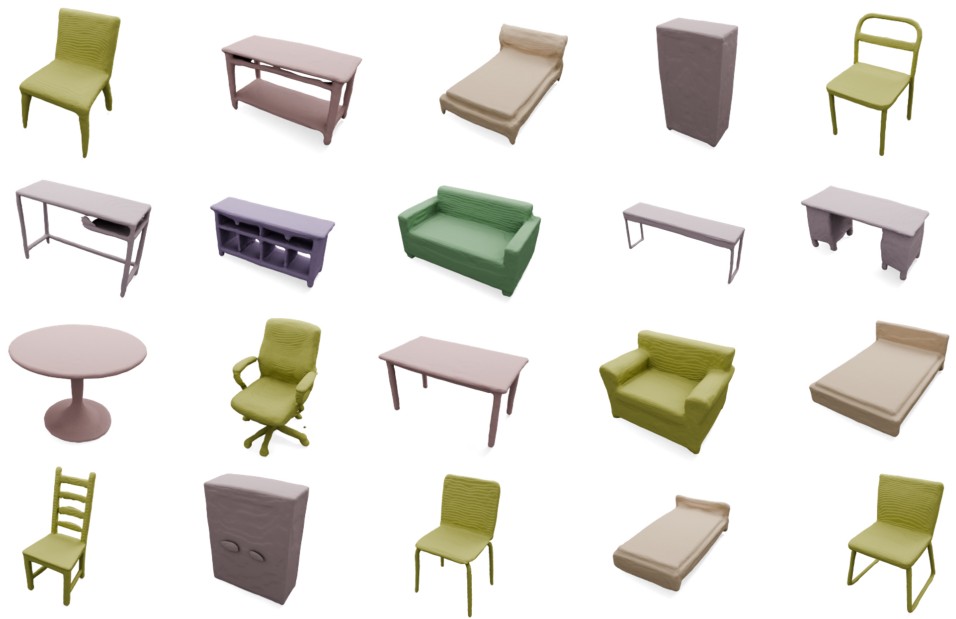

Figure 5: **Unconditional results.** Injecting $\varnothing$ as a condition to our conditional diffusion model, i.e., effectively disabling the conditioning mechanism, results in high-quality and diverse results.

forms the foundation for subsequent downstream tasks like mixed reality applications, our current model assumes a static scene geometry. Future work could integrate object affordance and articulation into our shape prior [34] to enable more immersive human-scene interactions.

**Broader Impact** We do not anticipate any societal consequences or negative ethical implications arising from our work. Our approach advances the holistic understanding of 2D perception and 3D modeling, benefiting various research areas.

# 5  Conclusion

In this paper, we present a novel diffusion-based approach for coherent 3D scene reconstructions from a single RGB image. Our method combines a simple yet powerful denoising formulation with a robust generative scene prior that learns inter-object relationships by exchanging relational information among all scene objects. To address the issue of missing ground-truth annotations in publicly available 3D datasets, we introduce a surface alignment loss $\mathcal{L}_{\text{align}}$ to jointly train shape and pose, effectively leveraging our shape representation. Our approach significantly enhances 3D scene understanding, outperforming current state-of-the-art methods across various benchmarks, with +12.04% $\text{AP}_{\text{3D}}^{15}$ on SUN RGB-D and +13.43% F-Score on Pix3D. Extensive experiments demonstrate that our contributions – 3D scene reconstruction as a conditional diffusion process, scene prior modeling, and joint shape-pose training enabled by $\mathcal{L}_{\text{align}}$ – collectively contribute to the overall performance gain. Additionally, we show that our model supports unconditional synthesis and generalizes well to other indoor datasets without further fine-tuning. We believe these advancements lay a solid foundation for future progress in holistic 3D scene understanding and open up exciting applications in mixed reality, content creation, and robotics.

# 6  Acknowledgements

This work was funded by the ERC Starting Grant Scan2CAD (804724) of Matthias Nießner and the ERC Starting Grant SpatialSem (101076253) of Angela Dai.

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

# A  Appendix

In the following, we show more qualitative results for scene reconstruction on SUN RGB-D [62] (Appendix B) and object reconstruction on Pix3D [64]. We provide detailed quantitative per-class comparisons supplementing the tables in the main paper (Appendix C). We additionally compare against a retrieval baseline on the ScanNet [11] dataset in Appendix D. Finally, we provide additional details on the architecture of our diffusion model in (Appendix E).

For a comprehensive overview of our approach and results, we encourage the reader to watch the supplemental video.

# B  Additional Qualitative Results

**Scene Reconstruction**    In Fig. 6, we show additional qualitative results of our method on test frames from SUN RGB-D. Despite strong occlusions and challenging viewing angles, our model predicts accurate scene reconstructions. Our generative scene prior learns common scene patterns, such as parallel object placements between the table and sofa or a bed and neighboring nightstands. In Fig. 8, we also demonstrate that our robust conditional scene prior can recover clean and matching shape reconstruction even for heavily occluded objects, *e.g.*, a chair for which only the back seat is barely visible.

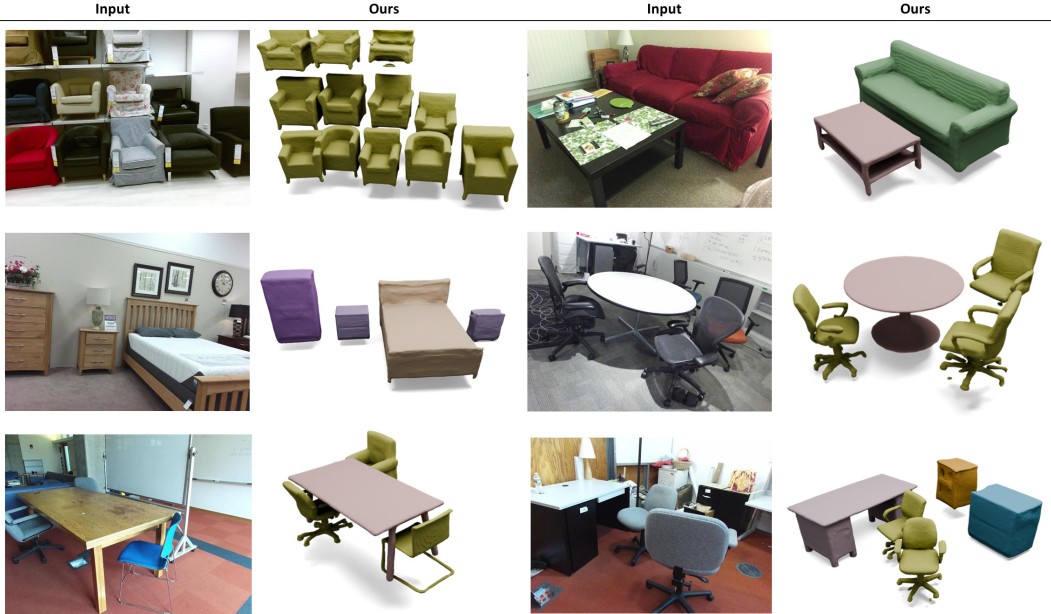

Figure 6: **Additional qualitative scene reconstruction results on SUN RGB [62].** Our diffusion-based scene layout and shape prediction approach achieves accurate results even for strongly occluded objects.

**Object Reconstruction & Unconditional Synthesis**    In Fig. 9, we show a qualitative comparison of single-view 3D object reconstruction on the Pix3D dataset. Unlike InstPIFu, which often produces noisy and incomplete surfaces, our image-condition diffusion model reconstructs clean and high-fidelity objects. Such a visual quality allows these reconstructions to be integrated into *e.g.*, mixed reality applications.

To probe the learned shape prior and investigate its shape synthesis capabilities, we input the 0-condition $\varnothing$ instead of extracted image features to our model. As shown in Fig. 5, our model learns a high-quality shape prior with fine details across various semantic classes.

| Input | Total3D | Im3D | Ours | Ground Truth |
|---|---|---|---|---|

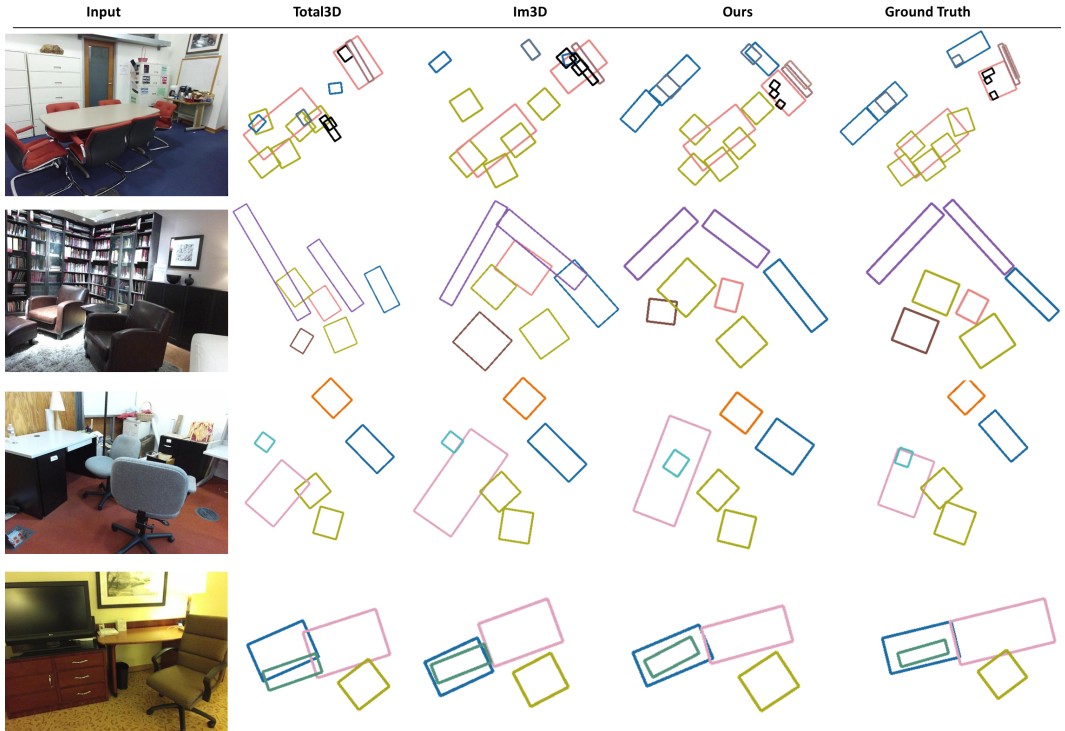

Figure 7: **Qualitative comparison of 3D pose estimation on the SUN RGB-D [62].** The input image is displayed on the left, and the predicted and ground-truth 3D arrangements are visualized as top-down orthographic views of the scene. We observe that Total3D frequently lacks a globally consistent structure, while Im3D predicts globally structured results but occasionally produces intersecting or floating objects. In contrast, our approach successfully recovers a coherent arrangement of objects within the scene by learning a robust scene prior.

## C Additional Quantitative Results

**Scene Reconstruction**     In Tab. 4, we show detailed comparisons of our approach against baseline methods, Total3D [48] and Im3D [77], on the 10 most common classes of SUN RGB-D. Our approach consistently outperforms all baseline methods on all classes except the "bed" class. We attribute this exception to the fact that beds are often only partially visible in the input view due to their spatial extent, which introduces higher variability. In contrast, Im3D employs a series of geometric losses and regularization terms, which seems to help in extreme amodal cases at the cost of additional loss balancing. Nevertheless, our method achieves a significant overall improvement of 12.04% in $AP_{3D}^{15}$ on these 10 classes, with particularly notable gains for "dressers" (+26.03%), "chairs" (+21.91%) and "cabinets" (+19.37%), showcasing the effect of our robust scene prior.

Tabs. 6 and 8 show the per-class comparisons and ablation studies on all 37 NYU classes in terms of $IoU_{3D}$ and $mAP_{3D}^{15}$. Our approach improves compared to Im3D by a +7.57% increase in $mAP_{3D}^{15}$ and +4.56% increase in class-mean $IoU_{3D}$ across all 37 classes. The ablation results highlight the importance of our diffusion formulation (+7.67% $mAP_{3D}^{15}$), scene prior modeling (+7.11% $mAP_{3D}^{15}$), and joint training using the surface alignment loss $\mathcal{L}_{align}$ (+0.72 $mAP_{3D}^{15}$).

**Object Reconstruction**     For single-view object reconstruction, we evaluate Chamfer Distance and F-Score on Pix3D and show per-class comparisons in Tabs. 7 and 9. Our image-conditional shape prior leads to significant improvements, +9.6% in Chamfer Distance and +13.43 in F-Score, while outperforming InstPIFu in most categories, except sofas and wardrobes in F-Score.

**Room Layout**     [48, 77] also predict the room bounding box with a separate network head. We study, how our model can also predict the room layout. For that we include the room bounding

box pose as part of the object poses during the diffusion process. We follow the room layout parameterization of [48, 77] and model the 3D room center directly instead of decomposing it as 2D offset & distance, which is done for the objects. In Tab. 3, we demonstrate that by denoising the pose of room layout, we outperform the regression-based methods.

Table 3: **Additional 3D room layout estimation on SUN RGB-D [62]**. We evaluate the 3D IoU of the orientied room bounding box. Our diffusion-based pose estimation lead to an improvement of +1.7% in Room Layout IoU.

|  | Layout IoU |
| --- | --- |
| Total3D [48] | 59.2 |
| Im3D [77] | 64.4 |
| Ours | **66.1** |

Table 4: **Additional per-class comparisons of 3D layout estimation on SUN RGB-D [62]**. Our method outperforms the baselines in most categories with overall strong improvements in **mAP$_{3D}$** evaluated at an IoU-threshold of 15%.

|  | bed | chair | sofa | table | desk | dresser | n.stand | sink | cabinet | lamp | mAP$_{3D}^{15}$ |
| --- | --- | --- | --- | --- | --- | --- | --- | --- | --- | --- | --- |
| Total3D [48] | 72.47 | 22.74 | 53.56 | 41.49 | 32.74 | 17.45 | 20.06 | 24.67 | 16.83 | 3.63 | 32.54 |
| Im3D [77] | **88.73** | 36.77 | 72.81 | 58.64 | 49.80 | 29.73 | 44.10 | 34.71 | 32.72 | 13.34 | 46.14 |
| Ours | 86.58 | **58.68** | **74.13** | **71.36** | **62.81** | **55.76** | **48.14** | **50.44** | **52.09** | **21.82** | **58.18** |

Table 5: **Quantitative comparison with ROCA [17] on the ScanNet dataset [11]**. While ROCA estimated each object's pose individually, our generative scene prior can reason about object relationships, leading to a +3.1% improvement in class-wise alignment accuracy.

|  | bathtub | bed | bin | b.shelf | cabinet | chair | display | sofa | table | cls. | inst. |
| --- | --- | --- | --- | --- | --- | --- | --- | --- | --- | --- | --- |
| ROCA [17] | 22.5 | 10.0 | **29.3** | 14.2 | 15.8 | **41.0** | **30.9** | 16.8 | 14.5 | 21.7 | 27.4 |
| Ours | **28.7** | **18.3** | 19.1 | **17.6** | **36.9** | 39.7 | 19.2 | **24.5** | **19.2** | **24.8** | **29.5** |

# D  Comparison to shape retrieval baseline on ScanNet

We compare with a shape retrieval baseline, namely ROCA [17]. Since ROCA requires full ground-truth supervision during training, we adopt their setup and train our model on the same 25,000 frames from the ScanNet [11] dataset with pose annotations derived from Scan2CAD [3], as well as the same CAD pool from ShapeNet [4]. We additionally adopt their full 9-DoF pose parameterization by predicting all 3 rotation angles. Following ROCA, we quantitatively evaluate the Alignment Accuracy in Tab. 5. Please refer to [3, 17] for the details of the evaluation. In Fig. 11, we can see that ROCA retrieves clean and complete shapes by definition. However, due to its limited shape database, it cannot capture all shape modes accurately, leading to shape mismatches. Our reconstruction-based approach instead can recover faithful shape results while simultaneously predicting a coherent object arrangement.

# E  Architecture Details

**Object Pose Parameterization: Normalization**     To ensure a reasonable signal-noise ratio [28] among the object pose parameters, we normalize the parameters to $[-1, 1]$ by dividing them by its max value and shift the range using a parameter-specific $\mu$ value. For this, we calculate the min-max ranges of all pose parameters, *i.e.*, rotation $\theta$, 3D scale $s$, and projected distance $\mathbf{d}$, within the train

Table 6: **3D pose estimation results** for all NYU-37 classes on SUN RGB-D [62]. We report the Average Precision (AP) at **15%** 3D-IoU threshold of the baseline and different variants of our approach: Our approach outperforms Total3D and Im3D on most semantic categories, especially on frequent classes likes chairs (+21.9%) or tables (+12.7%).

| | Total3D | Im3D | Ours | | | | |
|---|---|---|---|---|---|---|---|
| | | | no M2F | no diff. | no ISA | no joint | full |
| cabinet | 16.83 | 32.72 | 35.43 | 37.32 | 40.48 | 48.48 | **52.09** |
| bed | 72.47 | 88.73 | 76.23 | 84.58 | 86.50 | **90.71** | 86.58 |
| chair | 22.74 | 36.77 | 46.97 | 49.38 | 48.82 | 55.80 | **58.68** |
| sofa | 53.56 | 72.81 | 64.83 | 66.44 | 66.27 | 72.43 | **74.13** |
| table | 41.49 | 58.64 | 62.31 | 59.34 | 58.47 | 69.70 | **71.36** |
| door | 1.18 | 5.85 | 6.25 | 3.58 | 5.58 | **7.73** | 5.44 |
| window | 2.72 | 0.57 | 0.51 | **3.08** | 2.57 | 2.62 | 2.72 |
| bookshelf | 4.95 | 18.02 | 19.56 | 25.07 | 20.99 | **30.81** | 30.81 |
| picture | 1.21 | 1.66 | 0.99 | 2.04 | 1.31 | 1.80 | **3.95** |
| counter | 41.29 | 62.48 | 62.58 | 62.30 | 56.47 | 69.78 | **72.44** |
| blinds | 0.00 | 2.79 | 1.67 | 2.27 | 3.64 | 4.27 | **5.20** |
| desk | 32.74 | 49.80 | 52.31 | 48.78 | 48.93 | 60.20 | **62.81** |
| shelves | 9.72 | 18.16 | 14.58 | 16.31 | 14.51 | 25.31 | **28.01** |
| curtain | 1.30 | 7.69 | 9.19 | 3.94 | 6.76 | **11.93** | 10.43 |
| dresser | 17.45 | 29.73 | 36.07 | 41.86 | 50.91 | 53.06 | **55.76** |
| pillow | 9.41 | 19.48 | 19.37 | 23.10 | 20.54 | **33.45** | 28.99 |
| mirror | 0.50 | 0.84 | 4.22 | 1.11 | 2.04 | 8.15 | **9.98** |
| clothes | 0.00 | 0.00 | 0.0 | 0.00 | 0.00 | 0.00 | 0.0 |
| books | 4.23 | 7.16 | 5.42 | 11.26 | 10.73 | **17.18** | 12.76 |
| fridge | 25.00 | 40.47 | 27.13 | 42.66 | 37.59 | 45.90 | **46.17** |
| television | 10.88 | 14.49 | 13.89 | 11.95 | 10.71 | 19.81 | **23.55** |
| paper | 3.47 | 1.14 | 1.97 | 4.96 | 4.75 | 4.97 | **5.75** |
| towel | 4.35 | **14.80** | 2.68 | 8.11 | 8.19 | 11.02 | 12.99 |
| s.curtain | 0.00 | 0.00 | 0.00 | 0.00 | 0.00 | 0.00 | 0.00 |
| box | 7.40 | 11.52 | 15.86 | 17.43 | 17.72 | **29.02** | 24.42 |
| whiteboard | 1.40 | 2.59 | 2.68 | 1.66 | 3.17 | 4.18 | **5.44** |
| person | 22.12 | 19.22 | 38.32 | 31.48 | 28.45 | 55.10 | **56.39** |
| nightstand | 20.06 | 44.10 | 28.76 | 38.41 | 36.32 | 45.50 | **48.14** |
| toilet | 64.36 | **73.14** | 65.11 | 61.56 | 71.57 | 71.19 | 66.30 |
| sink | 24.67 | 34.71 | 30.49 | 32.01 | 39.60 | 42.94 | **50.44** |
| lamp | 3.63 | 13.34 | 12.90 | 12.88 | 12.48 | **21.84** | 21.82 |
| bathtub | 46.86 | 66.54 | 30.51 | 36.47 | 40.87 | 50.46 | **52.77** |
| bag | 13.67 | 8.45 | 8.66 | 13.78 | 16.52 | 18.89 | **21.69** |
| $\text{mAP}^{15}_{\text{3D (all)}}$ | 17.63 | 26.01 | 24.17 | 25.91 | 26.47 | 32.86 | **33.58** |
| $\text{mAP}^{15}_{\text{3D (10/37)}}$ | 30.56 | 46.14 | 44.63 | 47.10 | 48.88 | 56.07 | **58.18** |

set of SUN RGB-D. The 2D offsets to the 2D bounding box center are normalized by the image dimensions.

$$\mathbf{d} : \mu = 2.7, \max = 2.5, \tag{12}$$

$$\mathbf{s} : \mu = 3.5, \max = 7.0, \tag{13}$$

$$\theta : \mu = 0.0, \max = 3.14. \tag{14}$$

During training, the loss is computed on the un-normalized parameter ranges. After inference and for evaluation, we un-normalize each parameter according to its original range.

**Surface Alignment Loss: Point Sample Transformation** During training, for each object $o_i$, we use the predicted shape $\hat{\sigma}_i$ to estimate its scaffolding Gaussians $\hat{G}_j$. From each 3D Gaussian distribution, we directly draw 3D point samples $p_{(j,l)} \sim \mathcal{N}(\mu_j, \Sigma_j)$. This shape point cloud $P_i$ approximates the shape. With the predicted and un-normalized object pose $\hat{\rho}_i$, we define a 3D rigid transformation $\mathcal{R}^{4 \times 4}$ and transform the shape point cloud $P_i$ to the camera coordinate system. We use this transformed shape pointcloud $P_i^{\text{cam}}$ and the instance-segmented ground-truth depth map from

Table 7: **Per-class comparisons of shape reconstruction on Pix3D [64].** We report F-Score using the non-overlapping 3D model split from [37]. We observe noticeable improvements or comparable results on all categories.

|  | bed | b.case | chair | desk | misc | sofa | table | tool | w.robe | F-Score |
|---|---|---|---|---|---|---|---|---|---|---|
| Total3D [48] | 34.69 | 28.42 | 35.67 | 34.90 | 10.41 | 51.15 | 17.05 | 57.16 | 52.04 | 36.20 |
| Im3D [77] | 37.13 | 15.51 | 25.70 | 26.01 | 11.04 | 49.71 | 21.16 | 5.85 | 59.46 | 31.45 |
| InstPIFu [37] | 54.99 | 62.26 | 35.30 | 47.30 | 27.03 | **56.54** | 37.51 | 64.24 | **94.62** | 45.62 |
| Ours | **62.47** | **65.32** | **60.05** | **56.67** | **30.89** | 55.87 | **56.28** | **69.11** | 92.56 | **58.71** |

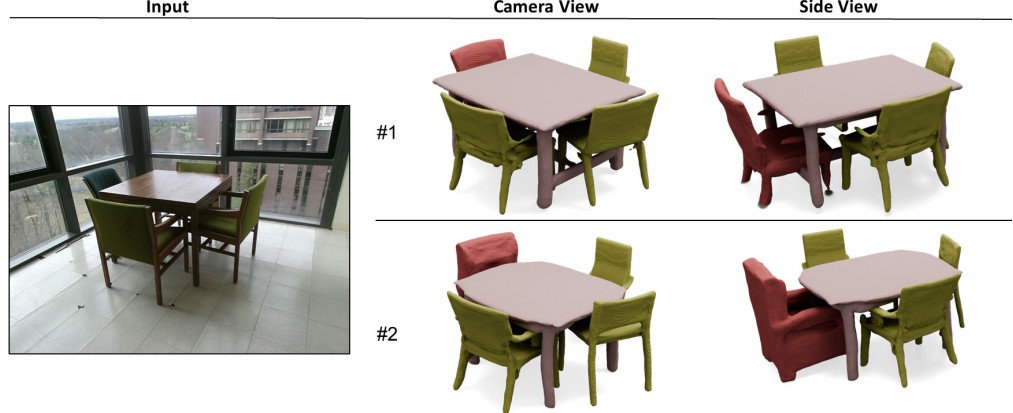

Figure 8: **Probabilistic behavior for partially occluded shapes.** In the input image, the left chair is heavily occluded, which allows for multiple plausible interpretations of the non-visible part of the shape. Our diffusion-based method derives faithful modes.

SUN RGB-D as the partial target pointcloud to measure the 1-sided Chamfer distance and to compute the surface alignment loss $\mathcal{L}_{\text{align}}$.

**Scene Prior Modeling: Inter-Object Relationships via Intra-Scene Attention** We use the multi-head attention mechanism [69] between the scene objects to allow them to attend to each other, effectively learning their inter-object relationships and the scene context. Specifically, given an unordered set $S = [o_1, o_2, ..., o_n], o_i \in \mathcal{R}^n$ per-object $n$-dimensional feature vectors, projection layers ($W^Q$, $W^K$ and $W^V$) and features $Q = S \times W^Q$, $K = S \times W^K$ and $V = S \times W^V$ after projection. we define the intra-scene attention as:

$$ISA(S) = softmax(\frac{QK^T}{\sqrt{d_d}})V \tag{15}$$

**Condition: Embedding Functions** After cropping the 2D image feature patch $\mathcal{R}^{W \times H \times C}$ from the frozen image backone $\Theta_{\text{I}}$, we apply adaptive average pooling to resize the per-object feature patches to a common 2D size leading to resized per-object feature crop of $8 \times 8$ and $C = 256$. This feature crop is further embedded using a small 2D CNN $\Theta_{\text{feat}}$ with 3 blocks of convolutional layers with 512 features, group norm, and leaky ReLU activation. The embedded feature crop is reshaped to a 4096-dim vector.

$\Theta_{\text{box}}$ is implemented as sinusoidal position encoding with 10 frequencies. This function is applied on a 2D bounding box, represented by the top-left and bottom-right corners, leading to an 84-dim vector per object. For $\Theta_{\text{cls}}$, we use a simple 1-hot encoding to embed the semantic class information. The final per-object condition information is the concatenation, resulting in a 4127-dim vector for each object.

**Reimplementation of SPAGHETTI [20]** Since the official code of SPAGHETTI does not include the training code and only provides checkpoints for two different shape classes (chairs,

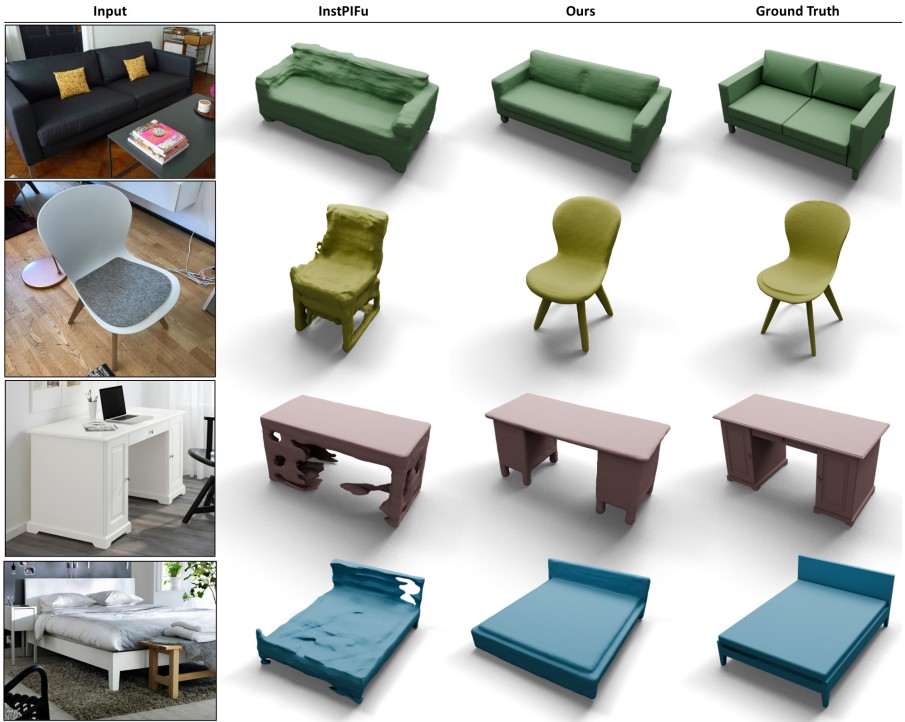

Figure 9: **Qualitative comparison of 3D shape reconstruction on the Pix3D [64].** While InstPIFu often produces noisy surfaces, our image-conditional 3D diffusion model synthesizes high-quality shapes that closely match the target geometries.

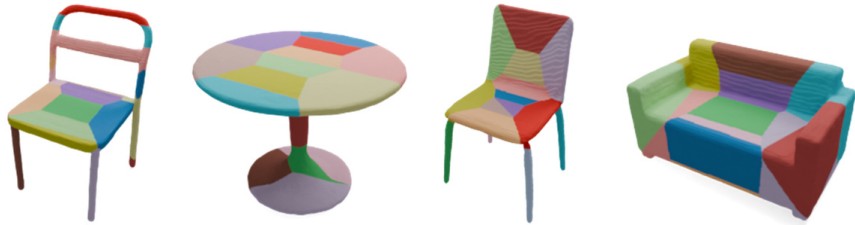

Figure 10: **Shape decomposition visualization.** We assign each vertex of the reconstructed mesh to the closest 3D Gaussian center and visualize the assignment with individual colors. Our scaffolding representation decomposes the shape into distinctive regions and aligns well with certain semantic parts, e.g., individual chair legs or the arm rests of a sofa.

airplanes), we re-implement the training procedure, loss function, and disentanglement loss following the description in the papers to train the full shape prior over all relevant shape categories. Random geometric augmentations are essential during training to achieve self-supervised disentanglement into extrinsic and intrinsic shape properties. We apply full 360-degree random rotations, uniform scale augmentation between 0.7 and 1.3, and translation jitter of $\mp 0.3$ on the disentangled extrinsic and target pointcloud. Further, we do not utilize the symmetry options of the original implementation.

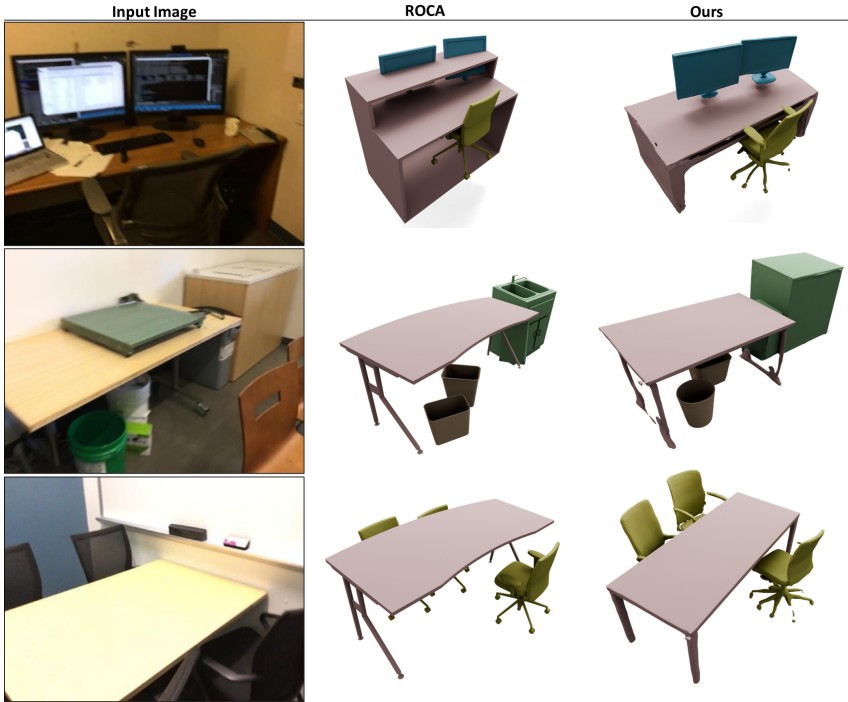

Figure 11: **Comparison with retrieval baseline method ROCA [17] on frames from ScanNet [11].** While ROCA cannot always retrieve a matching mode from the shape database, such as the desk in the first row, our diffusion-based reconstruction approach reconstructs accurate shapes and poses.

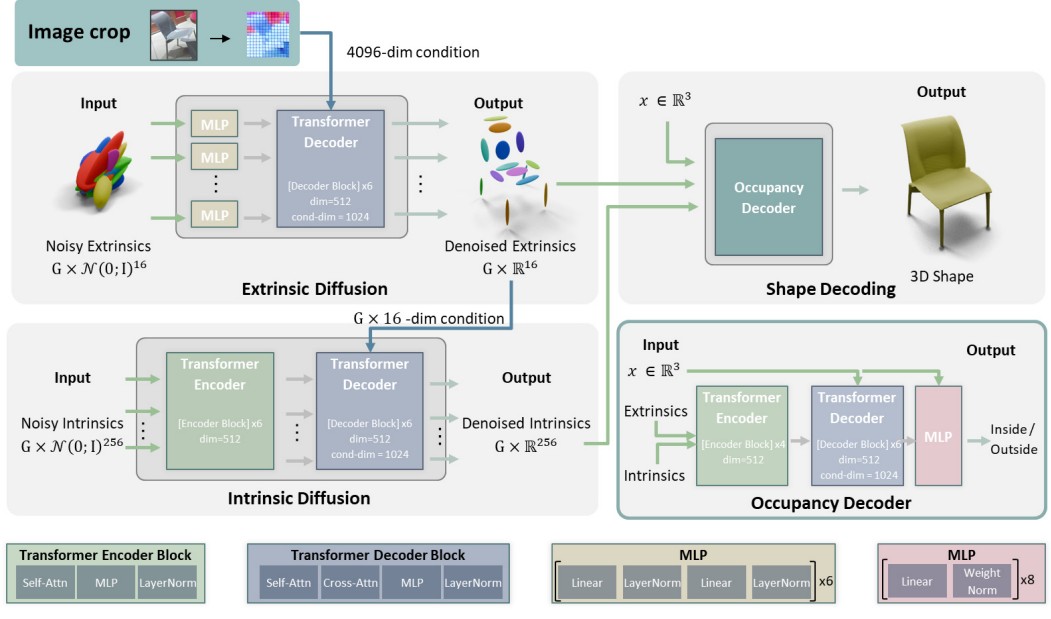

Figure 12: **Architecture Diagram of the Shape Diffusion Model.** The shape diffusion model consists of 3 sub-parts: An image-conditioned diffusion model, denoising the 3D Gaussians; a 3D Gaussian-conditioned diffusion model, denoising the intrisic vectors; and an Occupancy Decoder, which takes as input a 3D point coordinate and the denoised extrinsics & intrinsics and outputs an occupancy value indicating whether the 3D point is inside/outside of the shape.

Table 8: **Per-class pose estimation results** for all NYU-37 classes on SUN RGB-D [62]. We evaluate the pose estimation quality in terms of 3D IoU. Our scene prior formulation achieves improvements across all categories which particular high gains on common object classes like "chair" (+16.6%) or "desk" (+16.4%).

| | Total3D | Im3D | Ours | | | | |
|---|---|---|---|---|---|---|---|
| | | | no M2F | no diff. | no ISA | no joint | full |
| cabinet | 13.68 | 21.96 | 23.16 | 26.06 | 24.54 | **33.07** | 32.97 |
| bed | 32.28 | 42.65 | 41.53 | 48.98 | 44.87 | **52.67** | 52.25 |
| chair | 19.85 | 26.87 | 30.62 | 33.94 | 30.97 | 42.92 | **43.52** |
| sofa | 28.32 | 36.00 | 32.98 | 32.69 | 34.91 | 38.72 | **39.48** |
| table | 25.70 | 33.74 | 32.55 | 30.41 | 32.31 | **40.11** | 39.95 |
| door | 3.91 | 7.84 | 7.35 | 10.01 | 7.76 | **10.33** | 6.73 |
| window | 3.52 | 2.65 | 2.10 | 3.12 | 6.86 | 15.45 | **18.17** |
| bookshelf | 9.07 | 16.76 | 17.16 | 16.75 | 18.15 | **24.45** | 19.43 |
| picture | 2.35 | 5.30 | 4.69 | 5.70 | 4.32 | 3.36 | **6.32** |
| counter | 21.72 | 26.82 | 30.87 | 28.25 | 30.92 | **42.56** | 38.43 |
| blinds | 1.90 | 7.11 | 8.38 | 0.00 | 5.53 | 0.00 | 0.00 |
| desk | 21.09 | 28.21 | 28.12 | 34.57 | 27.51 | 44.22 | **44.68** |
| shelves | 10.33 | 14.92 | 14.01 | 16.35 | 14.81 | **24.32** | 17.60 |
| curtain | 5.09 | 9.46 | **10.40** | 7.99 | 9.39 | 2.55 | 0.00 |
| dresser | 16.84 | 23.29 | 23.08 | 22.86 | 27.82 | 29.19 | **32.56** |
| pillow | 11.07 | 17.65 | 16.62 | 18.12 | 16.77 | **19.05** | 16.69 |
| mirror | 2.05 | 4.45 | 5.65 | 4.83 | 4.11 | **9.03** | 5.81 |
| clothes | 0.00 | 0.00 | 0.00 | 0.00 | 0.00 | 0.00 | 0.00 |
| books | 6.81 | 8.97 | 9.59 | 14.00 | 15.63 | 12.30 | **20.48** |
| fridge | 18.41 | 27.02 | 19.92 | 16.18 | 24.61 | **26.85** | 23.36 |
| television | 9.59 | 14.11 | 12.74 | 12.60 | 11.62 | **19.73** | 18.93 |
| paper | 5.16 | 4.86 | 8.76 | 8.10 | **17.11** | 12.54 | 10.40 |
| towel | 7.46 | 10.53 | 7.26 | 10.83 | 8.32 | **18.79** | 13.71 |
| s.curtain | 33.12 | 13.49 | 30.53 | 0.00 | 0.00 | 0.00 | **9.41** |
| box | 9.40 | 12.04 | 16.18 | 17.91 | 16.47 | **24.55** | 23.82 |
| whiteboard | 4.06 | 6.27 | 5.94 | 4.07 | 6.39 | **6.46** | 5.47 |
| person | 24.14 | 23.33 | 28.94 | 15.40 | 21.89 | 19.50 | **28.91** |
| nightstand | 17.93 | 29.12 | 21.06 | 25.80 | 24.92 | **25.59** | 24.81 |
| toilet | 34.11 | 39.46 | 38.15 | 28.95 | 39.63 | **51.58** | 50.91 |
| sink | 19.92 | 25.40 | 21.50 | 20.54 | 24.99 | 20.81 | **26.60** |
| lamp | 9.63 | 15.90 | 12.92 | 13.94 | 13.20 | **24.33** | 24.22 |
| bathtub | 24.64 | 29.56 | 24.06 | 27.38 | 24.80 | 34.26 | **35.17** |
| bag | 11.18 | 11.70 | 13.63 | 18.41 | 16.38 | **22.74** | 21.60 |
| mIoU$_{3D}$ (all) | 14.15 | 18.10 | 18.53 | 17.30 | 18.76 | **22.79** | 22.66 |
| mIoU$_{3D}$ (10/37) | 20.52 | 28.31 | 26.75 | 28.98 | 28.82 | 35.16 | **36.10** |

Table 9: **Per-class comparisons of shape reconstruction on Pix3D [64].** We report Chamfer Distance using the non-overlapping 3D model split from [37]. Across most categories, our model achieves strong improvements compared to the baselines. Especially for frequent classes like "chair" or "table", we see a reduction of more than 45%.

| | bed | b.case | chair | desk | misc | sofa | table | tool | w.robe | CD |
|---|---|---|---|---|---|---|---|---|---|---|
| Total3D [48] | 22.91 | 36.61 | 56.47 | 33.95 | 137.50 | 9.27 | 81.19 | 94.70 | 10.43 | 44.32 |
| Im3D [77] | 11.88 | 29.61 | 40.01 | 65.36 | 144.06 | 10.54 | 146.13 | 29.63 | 4.88 | 51.31 |
| InstPIFu [37] | 10.90 | 7.55 | 32.44 | 22.09 | **47.31** | 8.13 | 45.82 | 10.29 | **1.29** | 24.65 |
| Ours | **8.43** | **7.11** | **17.63** | **19.81** | 65.29 | **8.41** | **21.06** | **8.07** | 2.01 | **15.05** |

