# OpenReview forum: "Coherent 3D Scene Diffusion From a Single RGB Image"
_NeurIPS.cc/2024/Conference — NeurIPS 2024 poster_

### Official Review · Reviewer_BXMX · 2024-06-29

**Soundness:** 3
**Presentation:** 3
**Contribution:** 3
**Rating:** 6
**Confidence:** 4

**Summary:**

The task is to perform 3D scene reconstruction from a single RGB views, such that they can output both scene object poses and geometries. Prior work cannot jointly output both poses and geometries, one design choice that is validated by this work to be effective. Method-wise, extending on SALAD, this work introduces a diffusion model that learns a generative scene prior that can capture contextual information across scene elements. To achieve joint learning, they introduce a alignment loss such that they only need to single view supervision. The author conducts extensive experiments to justify their design choices.

**Strengths:**

1. The introduction of the surface alignment loss (Sec 3.6) provides a nice solution to jointly improve object poses and geometry estimation. This kind of supervision only needs image level supervision (no need for 3D supervision).

2. The experiments on comparison and ablation is informative, which reasons well the key design decision made in this paper.

3. The qualitative demo is multifaceted, which makes it easier to evaluate their pose and geometry quality.

**Weaknesses:**

1. Minor concern is that the paper over-claim a bit on the contribution. Despite phrasing as 3D scene reconstruction (as is mentioned in teaser, abstract and intro) from single view, the proposed framework in the method actually only reconstruct poses and geometries of foreground objects in the image without the background part, e.g., ground or wall.

2. Global shape prior cannot always fit the observed object perfectly, with different local details, e.g., Fig.3 shelf. This could accumulate error in the downstream alignment.

3. Furniture kinds of 3D objects, e.g., those in ShapeNet, are simple, which usually is easier to get good shape reconstruction. How would the proposed framework work on more intricate 3D objects, e.g., plants, trees?

**Questions:**

1. Could you clarify on how the proposed framework resolve with scale ambiguity given the single view setup?

2. Writing needs more polish. For example, Equation 1 seems incorrect (feel free to clarify in the rebuttal). Based on the description in Line 121, RGB image I seems to be given, which indicates it is one item to be conditioned on. And in Line 150, missing parenthesis for (See Sec. 4.2

3. Could you clarify on how the gradient being back-propagated through point sampling (Line 190)? "We directly sample m = 1000 points p(j,l) ∼ N (μj , Σj ) per Gaussian Gˆi,j resulting in a shape point cloud Pi = {p(j,l)|j ∈ {1,...,g},l ∈ {1,...,m}}." In SALAD, it has another cascaded diffusion Transformer that condition on extrinsic and intrinsic parameters to get the part-level point cloud from Transformer. But I am not sure if the process is the same in this paper.

4. Could you clarify how the texture of the 3D objects are learned? The qualitative results show objects with color but there is no supervision on the color.

**Limitations:**

Yes.

---

> ### Author Rebuttal · Authors · 2024-08-07
>
> **1. Reconstructing background objects**
>
> Indeed, structural elements can be considered part of holistic scene reconstruction.
> Including structural elements in the scene prior can certainly be beneficial and can improve the coherence of the scene.
> We will extend our formulation with the structural elements of the scene and train a layout estimator to reconstruct them. We will report our findings in the final paper - thanks a lot for this suggestion!
>
>
>
> **2. Global shape prior**
>
> Thanks for the comment. While our model produces clean shape modes even under heavy occlusions
> compared to blurry or incomplete shapes of prior work, our learned shape prior does not necessarily match the input image perfectly.
> In our alignment loss formulation, we sample points from the 3D Gaussians, which represent the coarse shape of the object. Incorporating the per-Gaussian intrinsics, which encode local details, into the loss formulation is an interesting experiment.
>
>
>
> **3. Reconstruction of natural objects, e.g. plants, trees**
>
> Thanks for the interesting question. In Fig. 3, 5 and 9, we show that our geometry-based model is able to reconstruct certain fine details, such as individual braces at the back of a wooden chair or thin table legs. However, for natural, intricate objects such as plants, geometric approaches generally struggle to reconstruct very fine details. To study the limits of our approach, we will include a comparison with an increased number of 3D Gaussians in the shape model.
> To further mitigate this issue, our approach can reconstruct the geometry of the object, and additional fine details, such as individual leaves, can be further represented by local textures, as has been done in computer games, or with 3D Gaussian splats.
>
> **4. Resolving scale ambiguity**
>
> Since our model learns object poses and scales together with a strong scene prior, our model naturally reasons about common scene scales and hence is less prone to ambiguities of individual objects given the scene context.
>
>
>
> **5. RGB image input**
>
> Thanks for the comment. We will go through the manuscript thoroughly and improve the writing and fix minor typos.
> Eq. 1 defines the overall problem setup of reconstructing multiple objects from a single RGB image ($\mathbf{I}$, L121). Instead of conditioning on the input image directly, we use an image feature extractor $\Theta_i$ and instance segmentation on the input image to extract per-detection image features, which are then used to condition the diffusion model ($\text{feat}_i$, L135, Eq. 6).
>
> **6. Gradient backpropagation through point sampling**
>
> For sampling we use `torch.distributions.MultivariateNormal` function with the estimated per-Gaussian centers $\mu$ and covariance matrices $\Sigma$. While the sampling itself is not differentiable, we can compute the gradients on the sampled points.
> During joint training with the proposed efficient alignment loss, we omit the additional denoising of the intrinsic vectors and the costly rejection of point samples using the shape decoder. Hence, our point sampling formulation is different from the shape decoding of SPAGHETTI [1] or SALAD [2].
>
>
>
> **7. Textures of objects**
>
> Our model does not predict object colors or textures, however, this is an interesting future direction.
> For visual clarity, we color each object according to its semantic class - we will indicate this in the captions of the figures.
>
> **References**
> - [1] Hertz et al. - “SPAGHETTI: Editing Implicit Shapes Through Part Aware Generation”, SIGGRAPH (TOG) 2022
> - [2] Koo et al. - “SALAD: Part-Level Latent Diffusion for 3D Shape Generation and Manipulation”, ICCV 2023

---

> > ### Comment · Reviewer_BXMX · 2024-08-13
> >
> > Thanks for the detailed clarification. I retain my rating. Nice work!

---

### Official Review · Reviewer_scCV · 2024-07-08

**Soundness:** 3
**Presentation:** 3
**Contribution:** 3
**Rating:** 6
**Confidence:** 4

**Summary:**

This paper proposes a novel framework for 3D scene reconstruction from single images based on diffusion models. This framework jointly predicts 3D object poses and shapes with seperate diffusion models and captures global scene context with a standard mutli-head attention module. To train this model, the paper proposes a surface alignment loss based on one-sided Chamfer Distance, which makes their method able to utilize depth maps as supervision. The experimental results on Sun RGB-D, ScanNet, and Pix3D and numerical ablation studies demonstrate the efficacy of proposed method.

**Strengths:**

This paper shows its strengths from several aspects:
1. It proposes an overall solid method for 3D scene reconstruction from single images and deals with the ill-posed nature of this problem with powerful generative prior from diffusion models.
2. The proposed alignment loss makes the method able to utilize partial supervision from depth maps for model training.
3. This paper conducted comprehensive experiments in both scene and object reconstruction and achieve good results comparing to previous methods.
4. The presentation and writing are overall good and clear.

**Weaknesses:**

My concerns are mainly about the description of the method:
1. The model architecture is not clear enough in Section 3.7. It seems that each object pose, each object shape, and each Gaussian, are denoised using an independent diffusion model? Where is ISA added and how is it integrated with other modules? Are the inputs of ISA from Equation (7)?
2. In Figure 1, are the noisy shape Gaussians as in shape with guassian noise? Or are they initial estimation directly from image features? It's a bit confusing.
3. Are the diffusion models for shape and pose trained with full 3D supervision before joint training with alignment loss? If yes, what if there are only ground truth depth maps in training data?

Besides, the overall idea of 3D generation from features as in Equation (6) combined with global context via attention has been explored in previous related works, such as part-based 3D object reconstruction, e.g., He, Qian, et al. "Single Image 3D Object Estimation with Primitive Graph Networks." Proceedings of the 29th ACM International Conference on Multimedia. 2021. This should be included in related works.

**Questions:**

1. According to the alignment loss, the predicted Gaussians should be converged to centered on the surface of 3D objects? Otheriwse, the direct sampling won't be around the surface?
2. How do you deal with varying number of objects within a single image? How do you deal with false positives and false negatives from detection?
3. Have you tried other normalization methods besides Equation (12-14)? Does it affect the learning efficiency?
4. Does the disentanglement of pose and shape helps with learning convergence?
5. How is the proposed method comparing to other SOTAs in single object reconstruction? Can this method be applied to part-level object reconstruction?

**Limitations:**

The authors adequately addressed the limitations and I do not have more to add here.

---

> ### Author Rebuttal · Authors · 2024-08-07
>
> **1. Clarification of model architecture**
>
> Thank you for the feedback regarding the architecture description of our model in Section 3.7. We will improve the clarity of this section in the final paper. In the rebuttal PDF, we have included an architecture figure of the shape model (Fig. 4).
>
> In particular, the pose diffusion model is conditioned on all objects’ image features of a scene simultaneously (Eq. 6 & 7). This setup allows the model to learn a scene prior via ISA by exchanging relational information between the objects and to denoise the individual object poses.
> During joint training, the entire model takes as input all scene object features (Eq. 6 & 7), estimates the scene context implicitly via ISA, predicts the object poses and the intermediate shape Gaussians of each object. These Gaussians are transformed into world space using the specific object pose to compute the Alignment Loss (Fig. 2, right).
>
>
> **2. Clarification of “Noisy Shape Gaussians” in Figure 1**
>
> The “Noisy Shape Gaussians” in Figure 1 depict the anisotropic 3D Gaussians of 3 objects at the beginning of the denoising process. During the backward diffusion process, these 3D Gaussians get denoised to form the intermediate, scaffolding representation of each shape. We will clarify it in the figure and the caption.
>
>
> **3. 3D supervision**
>
> Yes, the pose and shape models are first trained with respective 3D supervision before joint fine-tuning.
> We follow the training scheme of previous works, Total3D [3], Im3D [4], and train the poses on SUN RGB-D [5] and shapes on Pix3D [6]. Hence, our approach learns marginal distributions for layout and shape first. We then fine-tune for the joint distribution by using the provided depth map from SUN RGB-D and the proposed alignment loss.
>
> We agree that training on depth data only poses an interesting and challenging problem setting. In order to directly learn the joint distribution, image-based representation methods, such as EG3D [7], could be employed to learn meaningful shape priors from RGB-(D) data. These could be combined with depth and pose estimation like NOCs [8]. We will address this in the limitation section.
>
>
>
> **4. Context-learning via attention**
>
> Thanks for pointing us to this relevant work, we will include it in the related work section.
>
> **5. Surface alignment of Gaussians**
>
> In our experiments, we observed that the intermediate 3D Gaussians usually align well with the object’s parts (see Fig. 1 and Fig. 2), leading to sampled points that are close to the surface.
> For future work, it would be interesting to integrate surfel-based representations [9, 10] as promising alternatives for binding Gaussians to the surface of the object (2DGS [11]).
>
>
>
> **6. Varying number of scene objects**
>
> Our model can in fact handle a varying number of objects in the scene, because the dot-product attention in our ISA formulation is agnostic to the number of objects during train and inference time. In our experiments, we have not experienced negative impacts of false positives/negatives from the image detection backbone, however it is possible to perform NMS on the reconstructed 3D shapes to further filter out false detections.
>
> **7. Object pose normalizations**
>
> During our experiments, we did not try different normalization factors for the object pose parameters. We will include a comparison with different, e.g., unnormalized, object poses in the final paper.
>
> **8. Convergence of disentangled of pose and shape**
>
> Thank you for the interesting question. Investigating the convergence properties of the presented disentangled pose and shape formulation is indeed a meaningful addition.
> We will perform a comparison in a controlled environment, e.g., using synthetic data with full 3D ground truth such as 3D-FRONT [12], and include it in the final paper.
>
>
> **9. Comparison with SOTA single-object methods**
>
> We show comparisons for single object reconstruction with InstPIFU [13] in Figure 8.
>
> **10. Can it be applied to part-level object reconstruction**
>
> In our experiments, the scaffolding 3D Gaussians align well with individual objects parts. For a part-level reconstruction, each vertex of the output shape mesh can be assigned to its nearest 3D Gaussian to get a part-level segmentation of the object. We will include a visualization of that in the final paper.
>
> **References**
>
> - [1] Hertz et al. - “SPAGHETTI: Editing Implicit Shapes Through Part Aware Generation”, SIGGRAPH (TOG) 2022
> - [2] Koo et al. - “SALAD: Part-Level Latent Diffusion for 3D Shape Generation and Manipulation”, ICCV 2023
> - [3] Nie et al. - “Total3DUnderstanding: Joint Layout, Object Pose and Mesh Reconstruction for Indoor Scenes from a Single Image”, CVPR 2020
> - [4] Zhang et al. - “Holistic 3D Scene Understanding from a Single Image with Implicit Representation”, CVPR 2021
> - [5] Sun et al. - “Pix3D: Dataset and Methods for Single-Image 3D Shape Modeling”, CVPR 2018
> - [6] Song et al. - “SUN RGB-D: A RGB-D Scene Understanding Benchmark Suite”, CVPR 2015
> - [7] Chan et al. - “EG3D: Efficient Geometry-aware 3D Generative Adversarial Networks”, CVPR 2022
> - [8] Wang et al. - “Normalized Object Coordinate Space for Category-Level 6D Object Pose and Size Estimation”, CVPR 2019
> - [9] Pfister et al. - “Surfels: Surface elements as rendering primitives”, SIGGRAPH 2000
> - [10] Zwicker et al. - “EWA volume splatting”, Visualization 2001
> - [11] Huang et al. - “2DGS: 2D Gaussian Splatting for Geometrically Accurate Radiance Fields”, SIGGRAPH 2024
> - [12] Fu et al. - “3D-FRONT: 3D Furnished Rooms with layOuts and semaNTics”, ICCV 2021
> - [13] Liu et al. - “Towards High-Fidelity Single-view Holistic Reconstruction of Indoor Scenes”, ECCV 2022

---

> > ### Comment · Reviewer_scCV · 2024-08-14
> >
> > Thank you for the response. The architecture diagram is very helpful and most of my concerns are resolved. I do believe this is a good work and would like keep my rating.

---

### Official Review · Reviewer_U6My · 2024-07-13

**Soundness:** 3
**Presentation:** 2
**Contribution:** 2
**Rating:** 5
**Confidence:** 2

**Summary:**

In this work, the authors propose a diffusion-based method for generating coherent 3D scenes from a single input image. They use a diffusion model to jointly denoise the 3D poses and geometries of all objects. To address the incomplete ground truth of existing datasets, they also introduce a surface alignment loss to better learn the 3D geometry. Experimental results show that the proposed approach significantly outperforms state-of-the-art methods.

**Strengths:**

- I like the formulation of this method. I think anisotropic 3D Gaussians are a good representation for 3D reconstruction. We could treat the 3D reconstruction in a coarse-to-fine manner and use the 3D Gaussians as the 3D proxy to guide the fine reconstruction.

- The results look promising. With partial ground truth for supervision, the proposed method learns complete geometry from a single RGB image with high occlusion.

**Weaknesses:**

- I am unclear on how to learn the shape information:

(i) For the shape code sigma_i (16 Gaussians x 16 dimensions per Gaussian), do you have the ground truth for it? If not, how do you apply the shape diffusion loss mentioned in Equation 10?

(ii) To decode the shape code from Gaussians to an occupancy grid, you apply a shape decoder diffusion model (mentioned in Line 213). How do you train this model? Do you include any details about this model?

- I am interested in how the proposed method learns the intermediate scene representation (i.e., anisotropic 3D Gaussians). You use the surface alignment loss as supervision. Are there other indirect supervisions? How do you ensure that the Gaussians do not intersect each other or become trivial (e.g., the same as each other)?

- If I understand correctly, you add inter-object attentions to the intra-scene attention to allow the pose and shape model to run in parallel. If so, this operator is quite common.

- I find the author may miss the citation and discussion with an important literature [a].

[a] CoReNet: Coherent 3D scene reconstruction from a single RGB image. ECCV 2020.

**Questions:**

- It is suggested to add notation for cls in Equation 6.

- It is suggested to add some figures for the ablation study.

- I find [b] to be quite related to this work. It is suggested to discuss this paper in the related work section. Combining some 3D architecture with the proposed method could be a nice extension.

[b] XCube: Large-Scale 3D Generative Modeling using Sparse Voxel Hierarchies. CVPR 2024.

**Limitations:**

I think the authors properly discuss the limitations of this work.

---

> ### Author Rebuttal · Authors · 2024-08-07
>
> **1. Clarification of shape learning**
>
> To support our latent diffusion approach for modeling shapes, we represent 3D shapes following the disentangled formulation of SPAGHETTI [1]. SPAGHETTI learns the disentanglement into 16 Gaussians and per-Gaussian “intrinsics” in a self-supervised way through augmentation with rigid transformations of the shapes. Since the official code only provides the inference part and checkpoints for chairs and airplanes, we re-implement the method according to the paper and train it across all categories on Pix3D [2]. From this trained model, we extract the shape codes $\sigma_i$ for supervision of the shape diffusion model.
> The shape decoder mentioned in L213 refers to the one of SPAGHETTI.
> We will add a more detailed description of our re-implementation and training in the paper as well as release the code and checkpoints together with the final paper.
>
>
> **2. Intermediate scene representation**
>
> The intermediate scene presentation consists of the per-shape 3D Gaussians from the shape diffusion model and the object pose from the object pose diffusion model. The 3D Gaussians are transformed to world space by the predicted object poses. Following baseline protocols and Sec. 4.2, we train our model in stages and use the respective supervision: The scene prior and object poses are trained on SUN RGB-D [3]; Shape reconstruction is trained on Pix3D [2] using the preprocessed 3D shape codes $\sigma_i$ (see above). The final model is then jointly trained on SUN RGB-D. We do not constrain the 3D Gaussians to be non-overlapping or  non-trivial, however by restricting the number of Gaussians (16 in our experiments), this encourages the shape model to distribute the Gaussians effectively in 3D space.
>
> **3. Intra-Scene Attention**
>
> Since we condition the model on all scene objects simultaneously, the intra-scene attention allows our model to learn a joint scene prior, e.g., properties such as object-object relationships and scene contexts. This is different from methods that individually regress the objects’ pose/shape (MeshRCNN [5], ROCA [6]) or refine the object poses in a second step (Im3D [4]).
>
> **4. Notion of cls in Eq. 6**
>
> ‘cls’ in Eq. 6 refers to the semantic class in L135. We will clarify this in the paper.
>
> **5. Figures for ablation studies**
>
> We included two additional figures (Fig. 1 & Fig. 2) for qualitative results of our ablation studies in the rebuttal PDF.
>
> **6. Additional related works**
>
> Thanks for pointing us to additional related works, we are happy to include and discuss them in the paper.
>
> **References**
>
> - [1] Hertz et al. - “SPAGHETTI: Editing Implicit Shapes Through Part Aware Generation”, SIGGRAPH (TOG) 2022
> - [2] Sun et al. - “Pix3D: Dataset and Methods for Single-Image 3D Shape Modeling”, CVPR 2018
> - [3] Song et al. - “SUN RGB-D: A RGB-D Scene Understanding Benchmark Suite”, CVPR 2015
> - [4] Zhang et al. - “Holistic 3D Scene Understanding from a Single Image with Implicit Representation”, CVPR 2021
> - [5] Gkioxari et al. - “Mesh R-CNN”, ICCV 2019
> - [6] Gümeli et al. - "ROCA: Robust cad model retrieval and alignment from a single image.", CVPR 2022

---

> > ### Comment · Reviewer_U6My · 2024-08-14
> >
> > Thanks for your rebuttal. I would like to keep my rate.

---

### Official Review · Reviewer_W8nX · 2024-07-13

**Soundness:** 3
**Presentation:** 2
**Contribution:** 2
**Rating:** 5
**Confidence:** 3

**Summary:**

This paper proposes an approach to reconstruct the 3D surfaces of multiple objects from a single RGB image. Given an instance-segmented RGB image as input, the poses and shapes of objects are jointly predicted and denoised using a diffusion model conditioned on the input image. A one-sided chamfer distance (from available ground truth surface point to sampled points in camera space) is used to provide supervision on incomplete surfaces. The output is geometrically evaluated against Total3D (3D detection-based composition), Im3D (implicit surface), and InstPIFu (pixel-aligned implicit surface) and shows improved scene coherence and shape quality.

**Strengths:**

+ Efficient training
+ Appropriate use of category-specific object shape priors
+ Clean ablation results

**Weaknesses:**

In order for the simple loss (one-sided chamfer distance) to work well, you need strong category-specific priors on generated shapes; it requires the estimated shape gaussians to be stable in the first place. The joint learning of poses is also inherently category-specific and might not generalize well to unseen or uncommon object categories. While this limitation is mentioned in the paper, I think this is an inherent limitation that is not easy to mitigate.

Limits of one-sided chamfer distance: In qualitative results containing office chair images for example, I noticed that many of them have extra leg parts hallucinated. I think this is an example of something that the one-sided chamfer loss won't be able to fix since the error isn't reflected in the loss. I'm surprised the method worked well despite this; this isn't necessarily a weakness, but tells us something about the problem we're trying to solve and evaluation methods (maybe "alignment is all you need"?). I think a retrieval-based baseline (with alignment) would have been interesting for the reasons mentioned.

I tend to think the improvements are mostly due to architectural advancements such as attention models (that can process arbitrary unordered variable-length objects) and better off-the-shelf shape models available. The paper appropriately applied those techniques and that is a good thing, but maybe it does not fundamentally improve our understanding of the problem. Perhaps this isn't a fair criticism considering the limitations and quality of other papers accepted in similar venues. I currently feel borderline about the paper. The holistic nature of the problem is difficult enough and the paper proposes one simple solution that I think can be used by other researchers still.

**Questions:**

(please see weaknesses section)

**Limitations:**

I think they are addressed well.

---

> ### Author Rebuttal · Authors · 2024-08-07
>
> **1. Generalization to unseen/uncommon object categories**
>
> Our shape prior is learned across all categories and in our experiments, we have seen that the shape Gaussians are indeed quite stable and align well with geometric parts (Figs. 1 & 2). However, we do rely on strong priors for seen categories, making generalization to unseen categories challenging. This limitation could be mitigated by training on larger shape datasets such as Objaverse [1].
>
> **2. Comparison to a retrieval baseline**
>
> We include qualitative results of the CAD retrieval method ROCA [2] in the rebuttal PDF (Fig. 3). While the retrieved CAD models are complete by definition, ROCA struggles to capture the correct mode of instances and produces misaligned and intersecting objects. We will include a detailed evaluation and comparison of this baseline in the final paper.
>
> **3. Improvements**
>
> In contrast to the regression-based baselines, such as Total3D [3], Im3D [4] and Mesh R-CNN [4], we introduce a fully probabilistic approach to model the joint distribution of object arrangements and shapes. As our quantitative and qualitative results show, our formulation leads to more accurate shapes and object arrangements. In particular, for ambiguous and challenging cases, our diffusion prior still yields plausible results where the baselines tend to produce inaccurate geometries and unrealistic poses.
>
> **References**
>
> - [1] Deitke et al. "Objaverse: A universe of annotated 3d objects.", CVPR 2023.
> - [2] Gümeli et al. - "ROCA: Robust cad model retrieval and alignment from a single image.", CVPR 2022
> - [3] Nie et al. - “Total3DUnderstanding: Joint Layout, Object Pose and Mesh Reconstruction for Indoor Scenes from a Single Image”, CVPR 2020
> - [4] Zhang et al. - “Holistic 3D Scene Understanding from a Single Image with Implicit Representation”, CVPR 2021
> - [5] Gkioxari et al. - “Mesh R-CNN”, ICCV 2019

---

> ### Comment · Reviewer_W8nX · 2024-08-13
>
> Thank you for the rebuttal. My "borderline accept" rating remains the same.
> I agree with other reviewers that the paper is above threshold.

---

### Author Rebuttal · Authors · 2024-08-07

We would like to thank the reviewers for their constructive and valuable feedback. We are pleased that *all* reviewers recognized the soundness of our approach for the challenging problem of single-image 3D reconstruction.

Our method was highlighted as solid (scCV) and efficient (W8nX) by appropriately employing strong diffusion priors (W8nX, scCV). The proposed surface alignment loss was well-received (scCV, U6My, BXMX), as was the use of anisotropic Gaussians for shape representation (U6My). These contributions led to results described as “multi-faceted” (BXMX), “good” (scCV) and  “promising” (U6My). Additionally, we are glad that our experimental setup and ablation studies were received as “comprehensive” (scCV), informative (BXMX) and “clean” (W8nX).

---

### Decision · Program_Chairs · 2024-09-25

**Decision:**

Accept (poster)

**Comment:**

The paper received four reviews from expert reviewers. The rebuttal has cleared up most remaining questions, and in the end, all reviewers retain their positive rating and recommend acceptance.
The reviewers highlight the novelty of the formulation, the clarity of the experiments and the writing of the paper.
The AC strongly encourages the authors to include all additional results, comparisons, and clarifications in the appropriate sections of the paper.